# Current Status and Future of Artificial Intelligence in MM Imaging: A Systematic Review

**DOI:** 10.3390/diagnostics13213372

**Published:** 2023-11-02

**Authors:** Ehsan Alipour, Atefe Pooyan, Firoozeh Shomal Zadeh, Azad Duke Darbandi, Pietro Andrea Bonaffini, Majid Chalian

**Affiliations:** 1Department of Radiology, Division of Musculoskeletal Imaging and Intervention, University of Washington, Seattle, WA 98195, USA; 2Department of Biomedical Informatics and Medical Education, University of Washington, Seattle, WA 98195, USA; 3Chicago Medical School, Rosalind Franklin University of Medicine and Science, North Chicago, IL 60064, USA; 4Department of Radiology, Papa Giovanni XXIII Hospital, 24127 Bergamo, Italy; 5School of Medicine, University Milano Bicocca, 20126 Milan, Italy

**Keywords:** multiple myeloma, radiology, artificial intelligence, machine learning, radiomics, segmentation

## Abstract

Artificial intelligence (AI) has attracted increasing attention as a tool for the detection and management of several medical conditions. Multiple myeloma (MM), a malignancy characterized by uncontrolled proliferation of plasma cells, is one of the most common hematologic malignancies, which relies on imaging for diagnosis and management. We aimed to review the current literature and trends in AI research of MM imaging. This study was performed according to the PRISMA guidelines. Three main concepts were used in the search algorithm, including “artificial intelligence” in “radiologic examinations” of patients with “multiple myeloma”. The algorithm was used to search the PubMed, Embase, and Web of Science databases. Articles were screened based on the inclusion and exclusion criteria. In the end, we used the checklist for Artificial Intelligence in Medical Imaging (CLAIM) criteria to evaluate the manuscripts. We provided the percentage of studies that were compliant with each criterion as a measure of the quality of AI research on MM. The initial search yielded 977 results. After reviewing them, 14 final studies were selected. The studies used a wide array of imaging modalities. Radiomics analysis and segmentation tasks were the most popular studies (10/14 studies). The common purposes of radiomics studies included the differentiation of MM bone lesions from other lesions and the prediction of relapse. The goal of the segmentation studies was to develop algorithms for the automatic segmentation of important structures in MM. Dice score was the most common assessment tool in segmentation studies, which ranged from 0.80 to 0.97. These studies show that imaging is a valuable data source for medical AI models and plays an even greater role in the management of MM.

## 1. Introduction

Multiple myeloma (MM) is a malignancy characterized by the uncontrolled proliferation of clonal plasma cells and the abnormal production of monoclonal immunoglobulin [1,2]. It is the second most common hematological malignancy following lymphoma and accounts for 0.9% of all cancer diagnoses [3,4]. Five-year survival rates of MM are estimated to be 74.8% and 52.9%, respectively [4]. MM symptoms can be summarized using the acronym CRAB: hypercalcemia, renal failure, anemia, and bone disease [5]. Other symptoms of the disease include weight loss, fatigue or general weakness, paresthesia, hepatomegaly, splenomegaly, and fever. Lytic bone lesions are present in 70–80% of patients at the time of diagnosis [6]. These lesions typically involve sites of the red bone marrow, with a prevalence of 49% in vertebral bodies, 35% in the skull, 34% in the pelvis, and 33% in the ribs [6,7]. The high prevalence of bone lesions in MM highlights the importance of imaging in the diagnosis and prognostication of MM [8]. The potential involvement of any bone segment highlights the need for whole-body techniques.

While a radiographic skeletal survey was recommended in the past, changes in diagnostic imaging for MM were made based on the revised diagnostic criteria for MM set by the International Myeloma Working Group (IMWG) in 2014 to account for newly discovered biomarkers [3]. This helped retire the previously used radiographic skeletal survey as an initial imaging test, mainly due to its high false-negative rate (30–70%), and helped introduce the use of new and more advanced modalities, including low-dose whole-body computed tomography (LDWBCT), whole-body magnetic resonance imaging (WB-MRI), and [7] 18F-fluorodeoxyglucose (FDG) positron emission tomography/CT (PET-CT) [8,9,10,11]. LDWBCT can be used as an initial diagnostic test since it is readily available and inexpensive. WB-MRI has a higher negative predictive value compared to LDWBCT and can be used to provide complementary information [2]. For post-treatment evaluation, WB-MRI and PET-CT are used since marrow signal intensity and FDG avidity changes occur before structural changes [3,12].

The introduction of these advanced imaging techniques for MM diagnosis highlights the need for a comprehensive understanding of the current and future roles of imaging in MM. The role of artificial intelligence (AI) in oncological imaging has been growing over the last decade and has been studied in other disease states [13,14,15]. For example, radiomics is a rapidly emerging research field, with several studies suggesting its potential to assist with the accuracy of disease diagnosis as well as the estimation of survival [14,16,17,18]. Previous systematic reviews of various oncological diseases have also provided insight into the progression of AI by highlighting challenges with validity [13,19]. However, the assessment of AI, including the validation of radiomics and segmentation in MM, has not yet been reviewed.

The purpose of this systematic review is to assess the status of AI in MM imaging and provide a future avenue for researchers and radiologists. It is noteworthy to mention that while a review by Allegra et al. discusses the applications of artificial intelligence in MM in general, that review is mostly focused on studies that use other types of data, including clinical data, and only briefly touches upon imaging in MM [20]. In contrast, our work is only focused on imaging in MM.

## 2. Materials and Methods

Preferred Reporting Items for Systematic Reviews and Meta-analysis (PRISMA) guidelines were used to conduct this systematic review [21], this research was not registered under PRISMA.

### 2.1. Search Strategy

We conducted a systematic literature search using PubMed (Medline), Embase, and ISI Web of Science to extract eligible studies from 2000 to April 2022. Search strategies focused on three main concepts of the use of AI (using “machine learning”, “deep learning”, “artificial intelligence”, “automated”) in radiologic examinations (using the following terms: “imaging”, “image”, “MRI”, “CT”, “PET”, “radiograph”, “radiographic”, “diagnostic”) in patients with MM. No limitations were applied. The results of our search findings are summarized in Figure 1.

### 2.2. Study Selection

Original studies that assessed the role of machine learning in the diagnosis, segmentation, or interpretation of all imaging types of MM were considered eligible for inclusion. The exclusion criteria consisted of review or commentary articles and studies with animal or cadaver subjects. The titles and abstracts of all obtained studies were independently screened by two reviewers (E.A. and F.S., postdoctoral research fellows with 2 years of research experience in radiology). After the exclusion of the duplicate studies, the full text of all eligible articles was assessed. All discrepancies were addressed, and a mutual consensus was reached among the authors regarding the final inclusion. 

### 2.3. Data Extraction

Authors’ names, years, and descriptive data of all studies including sample size, study design, imaging modality, techniques, parameters, reference standard, and the subject matter of each study were extracted. The following characteristic data were also obtained if they were provided: feature reduction strategies, which are often used in radiomics studies to prevent overfitting; the analysis tool that was used for the project, including methods like ridge regression, LASSO, XGBoost, and deep learning; the performance measures, including the area under the receiver operator curve (AUROC), accuracy, sensitivity, specificity, number of readers, the portion of the sample size used for training a model, conclusion, and pros and cons of each paper.

### 2.4. Study Evaluation

We used the Checklist for Artificial Intelligence in Medical Imaging (CLAIM) criteria, first introduced by Mongan et al., as a checklist for the quality of AI-related studies in radiology to evaluate each manuscript [22]. The checklist has 42 components, 4 for the title, abstract, and introduction; 28 for methods, 5 for results; and 5 for discussion and other necessary information. We evaluated each component and gave a score of 1 if the study was compliant with that criterion and a score of 0 otherwise. A physician with 4 years of experience in medical AI research reviewed each paper to determine compliance with each of the criteria. Compliance was defined based on the descriptions provided in the original CLAIM paper. For example, for the “How missing data were handled” criteria, an article was considered compliant if the authors stated that they did not have any missing data or if they described the strategy they used to deal with missing data. We calculated the final score by adding all the points together. In addition, we reported section-specific scores as well. We could only analyze studies that were full-length manuscripts and worked on developing an AI model and testing it.

## 3. Results

### 3.1. Study Selection

Our search yielded 977 results. After we removed the duplicates and screened the remaining studies based on titles and abstracts, 14 relevant articles were selected. Articles were commonly excluded because they were about other diseases, used pathology slide images or flow cytometry data, used other clinical data without the use of imaging, or used data acquired from imaging performed on cadavers or animals. Figure 1 shows the PRISMA flow diagram. Out of these 14 studies, 4 were abstracts that were presented at conferences, but for the sake of comprehensiveness, we included them in this paper.

### 3.2. Characteristics of Included Studies

Out of the nine articles and four abstracts included, 6/14 (43%) used CT images [23,24,25,26,27,28], 3/14 (21%) used PET-CT scans [29,30,31], and 5/14 (36%) used MRI as their imaging modality [17,32,33,34]. A total of 5/14 (36%) studies used WB imaging or imaging from affected body parts, 1/14 (%) focused on the femur bone, 1/14 (7%) focused on the spine and pelvic bones, and 3/14 (%) focused on the spine alone. Meanwhile, 5/14 (36%) studies used radiomics [17,23,32,34,35], 6/14 (%) (including two that focused on radiomics and one that involved histogram analysis) focused on segmentation of regions of interest (ROI), like bone marrow or bone structures, 1/14 (7%) used bone subtraction maps to assist radiologists, 1/14 (7%) used cumulative CT scores to predict disease severity, and 2/14 (14%) used histogram analysis of CT values in the bone marrow to predict bone marrow infiltration levels. An overview of these articles and their findings are summarized in Table 1 and Table 2.

### 3.3. Radiomics Studies

Radiomics was one of the main focuses of studies looking into AI applications in MM imaging. Radiomics is a set of quantitative features extracted from medical images that have proven useful in predicting disease features and outcomes [36]. A typical pipeline for a radiomics study includes acquiring a set of images (with the same scanner and protocol), normalization of those images using bias correction and normalization methods like z-score normalization, in addition to resampling the images so that each pixel for all the images has the same size and images are uniform. ROI segmentation can be manually performed by a radiologist or automatically using AI models to delineate the actual ROI (usually bone marrow or bone lesions in MM), calculating radiomics features, which is often conducted using packages like Pyradiomics, selecting the most relevant ones using feature selection strategies like LASSO, and finally performing the analysis using machine learning techniques like ridge regression, decision trees, or deep learning [36]. External validation was used to assess the models’ performance (Figure 2). In MM, radiomics was used to predict a variety of factors, including relapse, differentiation of bone lesions from metastasis, high-risk cytogenic abnormalities, and plasma cell infiltration levels. One study focused on CT scans, whereas the other four focused on MRI [23].

Schenone et al. used radiomics to detect relapse in a retrospective cohort of 33 patients with MM who underwent routine CT follow-ups [23]. Of these, 17 patients relapsed, as indicated by their clinical records. They extracted 109 radiomics features from lesions on baseline WB-CT. Principal component analysis was used and three different strategies were applied to choose the most relevant features. Fuzzy c-clustering was used to predict relapse, and Hough transformation (HTF) was used to divide the data into two clusters. Data bootstrapping was applied to generate confidence intervals. Out of the two clusters, the one that contained more relapse cases was considered the relapse cluster, and the other one was the not-relapse cluster. The Critical Success Index (CSI) was calculated by dividing the number of true positive (TP) predictions by the number of TP, false positives (FP), and false negatives (FN). These findings were compared with those of the cytogenetics test. The best-performing model was the HTF model on the dataset of features, which correlated with Bone-GUI and achieved a CSI of 0.52 (±0.1), whereas the CSI for the cytogenetics test was 0.44 (±0.16).

Another study by Xiong et al. tested radiomics features to differentiate MM bone lesions of the spine from other metastatic lesions of cancers [34]. Conventional MRI sequences were used, including T1 and T2 weighted sequences for this predictive model analysis. The gold standard was clinical diagnosis, with or without biopsy. A sample of 107 patients (60 patients with MM and 47 with metastatic lesions) was included, and the lesions were manually segmented on MRI sequences. To choose the most relevant features, LASSO regression was used and a threshold for minimum intraclass correlation was set. Finally, five different models including Random Forests, Support Vector Machines, Naïve Bayes (NB), K-Nearest Neighbor (KNN), and Artificial Neural Networks (ANN) were built to make predictions. To choose the best-performing model, cross-validation was used in the training set and Matthew’s correlation coefficient (MCC) was used as the measure of prediction accuracy. The best-performing model was ANN. The ANN on T2WI had an MCC of 0.605 in predicting metastasis vs. MM lesions in their validation cohort. In addition, the model was trained to predict the metastasis subtypes but had lower accuracies.

Liu et al. conducted a similar study to predict high-risk cytogenic abnormalities using conventional MRI sequences [32]. High-risk cytogenic abnormalities are diagnosed using Fluorescence in situ hybridization (FISH) as the gold standard. A retrospective cohort of 50 patients with MM for whom MRI and FISH tests were available was included. Lesion segmentation was performed manually. Intraclass correlation and the SelectKBest method were used as the primary steps of feature selection. Then, LASSO was used to choose the final set of features (nine features). Logistic regression was used as a predictive model, and the area under the curve (AUC) was used as a measure of performance. The model was run using both a radiomics-only dataset and a combined dataset that included age and sex. The best-performing model, which was logistic regression on the combined dataset, achieved an AUC of 0.87 on the test set.

Li et al. conducted a study on estimating the overall survival in patients with MM using a combined set of radiomics and clinical features [17]. They also compared their findings with other risk models for MM, including the international staging system model and the Durie–Salmon Staging System. Their sample consisted of 121 patients with MM who underwent lumbar MRI. A radiologist segmented out the L1-L5 lumbar vertebra. After preprocessing, 1136 radiomics features were extracted from each vertebra. For each patient, the mean value of each feature was calculated for the five lumbar vertebrae. Various feature selection techniques were used, including univariate Cox models, Spearman’s correlation, and LASSO Cox regression. Using a linear model, they generated ‘rad-scores’ for each patient based on the radiomics features. A tool called the X-Tile plot was used to set a threshold of rad-score for high-risk and low-risk patients. The clinical features that correlated with the overall survival were also included to create a nomogram. The clinical features included a beta-2 macroglobulin of more than 5.5 mg/L, 1q21 gain, and del(17p) mutations. Their best model, the radiomics nomogram, achieved a C-Index of 0.81 in the validation cohort.

Finally, an abstract by Wennman et al. investigated the creation of a pipeline for the automatic calculation of the percentage of plasma cell infiltration in the bone marrow [35]. Their research consisted of two stages. In stage one, data from 541 MRIs of 270 patients with MM were used to develop an automatic segmentation tool for the pelvic bone marrow. Two radiologists manually segmented the MRIs. An nnU-Net architecture was used for the neural network development. The tool achieved a mean Dice score of 94% on the test set. In stage two, radiomics features extracted from the automatically segmented bone marrow were used to predict the percentage of plasma cell infiltration. A random forest classifier was used for the prediction task. Their tool achieved a mean absolute error of 14.3 compared to biopsy results. For comparison, they asked two radiologists to rank plasma cell infiltration levels on the training and test sets into three levels: none to mild, moderate, and severe. Then, the mean plasma cell infiltration percentage was used for each category in the training set to predict the percentage of plasma cell infiltration in the test set based on the radiologists’ categorizations. The mean absolute error for this prediction task was 16.1. The authors concluded that their tool has comparable accuracy to that of radiologists in predicting the percentage of marrow plasma cell infiltration.

### 3.4. Segmentation Studies

Another body of research was dedicated to the development of algorithms for the automatic segmentation of body structures that are important in MM like bone marrow or individual bones. Some of these studies did not specifically target patients with MM but mentioned that their findings could be used in MM patients. Segmentation includes the use of an algorithm, like a deep learning model, or setting threshold levels to define the boundaries of a region of interest on a medical image. Automatic segmentation was part of many of the studies we reviewed. Two studies examined segmentation algorithms based on MRI images, one on PET images, and four on CT images. One of the studies on CT images transferred the generated mask to concurrent PET images.

Fraenzle and Bendl [24] used a threshold model to define all the bone regions in the skeleton on CT images. Then, they used a flood-fill algorithm to fill the bone regions and create another mask. The difference between the two masks was calculated and considered to be the bone marrow regions. To assign each bone marrow region to a bone structure, PCA and a random forest classifier were used. They showed that this model can effectively categorize each bone marrow region into their respective bone structures on axial leg CT images.

A study by Shi et al. [31] examined the segmentation of bone lesions in 12 patients with MM using PET-CT imaging. They used and compared a V-net-based neural network and a W-net-based neural network developed using 70 phantom images generated by the researchers. A Dice score of 89.3% was obtained for the segmentation task on the test set. These results were higher than those of other machine learning algorithms like random forests, k-nearest neighbors (kNN) classifier, and support vector machines (SVM). 

Another study by Takahashi et al. [30] used automatic segmentation of bones on PET-CT images to calculate PET quantitative parameters of bone involvement in MM patients. Segmentation was performed on the CT images using a global threshold of Hounsfield unit values to generate a mask. Morphological closing, which is a procedure that fills gaps, was performed to account for the soft portions of the skeleton. The mask was transferred to the PET images, and the maximum standardized uptake value (SUV), mean SUV, and standard deviation of SUV were calculated for all bone structures, except for the skull. The article did not discuss the accuracy of their segmentation model. However, their predictions based on SUVmean at the level of bone involvement correlated with the results of a visual assessment performed by a nuclear medicine specialist. The study concluded by mentioning that their pipeline provides a standardized method to assess bone involvement in MM patients.

Wennman et al. [33] conducted a research project on developing a segmentation algorithm to segment bone marrows at different anatomic locations. They used MRIs of 66 patients with smoldering MM. A radiologist manually segmented the bone marrow regions labeling 30 different compartments, including the left and right femur, hip, sacrum, and humerus, in addition to C2-C7, T1-T12, and L1-L5. The nnU-Net Convolutional neural network was trained on the training dataset. On test data (14 patients) the model achieved mean Dice scores of 0.80–0.97 on all compartments. 

### 3.5. Other Types of Research

There were instances of other uses of AI in MM imaging. These included the use of bone subtraction maps [27], cumulative CT scores [26], histogram analysis of bone marrow [28,29], and calculation of bone marrow infiltration using deep learning.

Horger et al. [27] performed an experiment to assess how bone subtraction maps can help radiologists monitor the course of bone disease in MM patients more accurately and efficiently using low-dose multidetector CT images. A retrospective sample of 82 patients with 188 low-dose WB-CT images was included. While the gold-standard tests for the detection of progressive disease were radiologic assessment and hematologic follow-up, they compared readings using bone subtraction maps with standard images. The authors showed that bone subtraction maps increased the accuracy of diagnosis by changing the diagnosis in 9.7% of the cases. In addition, bone subtraction maps helped reduce the scan reading time by about 25% compared to the standard method.

Fervers et al. [28] assessed the performance of an automated pipeline to measure bone marrow infiltration in a cohort of 35 patients with MM, monoclonal gammopathy of undetermined significance (MGUS), smoldering MM with WB-CT scans, and a concurrent bone marrow biopsy. A pre-trained neural network was used to segment the spine. Although the model was developed on healthy individuals, it properly left out bone lesions in MM patients. They used Hounsfield thresholds to separate the bone marrow from the cortical bone. Then, using histogram analysis of CT values, they calculated the amount of non-fatty bone marrow tissue. Using multivariate regression analysis, they showed that these values correlate with bone marrow infiltration (*p* < 0.007, r: 0.46) and can detect cases with lytic bone lesions to some extent. They concluded by stating that the automated pipeline can help reduce the number of patients undergoing invasive bone marrow biopsies to assess bone marrow infiltration.

Martínez-Martínez et al. used a similar strategy to detect bone marrow infiltration in MM patients [25]. Their sample consisted of 74 MM or MGUS patients and 53 healthy individuals. They used thresholding to select the femur bones. Two radiologists divided the patients into two groups: those with infiltration and those without infiltration. They used healthy individuals to generate a density model. They assumed that those with infiltration would be outliers in this model. They used a classifier (k-NN or soft-margin SVM) on a set of two features extracted from a previous model. ROC curves were used to determine the classifier parameters. Their best model was the one that aimed to distinguish healthy individuals from those that had infiltration using SVM, achieving an AUC of 0.995 (±0.017) on the test data. Their third experiment, which was aimed at distinguishing those with bone marrow infiltration from other MM or MGUS patients, achieved an AUC of 0.894 (±0.070).

Another related area of research in this field is the determination of normal bone marrow characteristics using imaging. Satoh et al. [29] analyzed PET-CT images of 98 healthy individuals who underwent imaging for screening purposes. They used a commercial tool that used a three-dimensional fully automated convolutional neural network to segment bone regions in the spine and pelvis based on CT images. After manual corrections and preprocessing to remove cortical bone, they used a mask to select bone marrow regions on PET images. Afterwards, histogram analysis was performed to extract the features of the normal bone marrow on PET images. The mean and maximum values of the SUV were calculated and corrected using lean body mass (abbreviated as SLU) in addition to entropy. The mean SLU was 0.79 (95% CI 0.78–0.90) in men and 0.75 (95% CI 0.74–0.76) in women in this study. In addition to calculating these values in their sample of healthy individuals, they showed that the mean SLU and entropy correlate inversely with age in both sexes. 

Finally, a study by Nishida et al. [26] showed that cumulative CT values (cCTv) of the bone marrow in patients with MM are correlated with disease severity and prognosis. A CT post-processing computer software (MABLE) was used to extract information. Hounsfield thresholds were used to detect the bone and bone marrow regions. A cohort of 91 patients with MM were included in addition to 36 patients with smoldering MM and MGUS for comparison. The diagnosis was based on the International Myeloma Working Group criteria. The Durie–Salmon stage, ISS, and R-ISS Staging criteria were used to stage MM patients. Their pipeline using the MABLE software automatically calculated cCTvs and they showed that these values are correlated with the diagnosis of MM and a higher stage of the disease. In addition, the authors showed that the administration of therapy reduces cCTvs.

### 3.6. CLAIM Checklist Evaluation

We analyzed eight of the studies discussed above using the CLAIM checklist. The scores ranged from 24 to 33 with an average of 26 and a standard deviation of 2.9. Compliance levels were lowest in the Section 2 and Section 3 with 59% and 52.5% compliance percentages, respectively. In the Section 2, the most overlooked components included “De-identification methods”, “Intended sample size and how it was determined” and “Initialization of model parameters”. In the Section 3, the most notable overlooked component was “Failure analysis of incorrectly classified cases”. Finally, a few studies shared links to their full study protocol and provided publicly accessible data or codes. Table 3 includes the CLAIM checklist, in addition to the percentage of studies that fulfilled this requirement.

## 4. Discussion

In this review, we investigated the trends in AI research of MM. While the studies were diverse in terms of methods and outcomes, most fell into two major categories: radiomics and segmentation. A lot of effort is being put into using radiomics to develop predictive algorithms for MM. These studies usually focused on predictive tasks that a human radiologist cannot perform, like detecting high-risk cytogenic abnormalities [32]. Most of the radiomics studies used intraclass correlations in addition to LASSO to select the most appropriate features [32,34]. The analysis methods were also diverse and included logistic regression [32], neural networks [34], random forests [35], and clustering methods [23]. While some studies used bootstrapping methods to generate confidence intervals for their results, others simply reported their performance measures as a single number.

### 4.1. Radiomics in MM

Radiomics has shown some promising results when using data hidden in imaging modalities to perform high-level prediction tasks. However, it has not yet reached a performance level that can justify its use in routine clinical settings [37]. In addition, there seems to be a lot of variation in how radiomics research is performed and how their performance is assessed [36,37]. Some of the results should be looked at with a grain of salt, as studies suffer from methodological errors. For example, in a study by Liu et al. [32], researchers used each lesion as a data point due to the low number of cases (each patient might have many lesions). In machine learning, data points, especially across the training and testing datasets, must be independent of each other; Which is not the case in this study and partly explains the high accuracy they achieved on their test set. In addition, the measures of performance in these studies were as diverse as their topics, making it hard to compare. Measures like CSI [23], MCC [34], and AUC [32] were used for studies with binary outcomes, whereas one study used the mean standard error (MSE) as a measure of performance. This diversity can be a result of researchers potentially choosing the measure that gives the highest score to their specific study. Nevertheless, the reviewed studies show that radiomics features can be used to predict disease features that previously required specific molecular or laboratory tests. The development of larger MM datasets of patients and advances in radiomics best practice guidelines may improve the performance of these models in the future.

### 4.2. ROI Segmentation in MM

The second largest category contained studies that focused on automatically segmenting regions of interest on medical images. Segmentation in radiology is the process of categorizing each pixel or voxel into a medical image to a specific anatomical or pathological class, such as identifying and labeling different structures or regions within the image, for diagnostic or analytical purposes. These studies used two major approaches for their tasks. The first approach used rule-based algorithms to segment regions of interest based on a threshold for their signal intensity or density [30]. An example could be the segmentation of bone by selecting regions of the image with Hounsfield values equal to or higher than that of the bone. Studies that used this technique did not provide any measure of accuracy, and segmentation was only a part of the study and was followed up by some other analyses. The second approach was to use state-of-the-art machine learning models like U-Nets or V-nets [28,31]. These are neural network structures that have shown to be effective in segmentation tasks. 

While radiomics studies are in the preliminary stages of development, segmentation algorithms seem to perform efficiently, often competing with the performance of a human radiologist. State-of-the-art neural network structures like U-Nets achieved Dice scores compared to human segmentations that were comparable to Dice scores between two human radiologists. Interestingly, in a study by Fervers et al. [28] the neural network previously developed for spine segmentation in healthy individuals performed well in segmenting the spine in patients with MM who had lytic bone lesions. As shown in some of the studies reviewed here, these segmentation tools can play an important role in developing automatic diagnostic or predictive algorithms in the future. The study by Horger et al. was probably the only study that assessed the clinical applicability of an AI tool and gave us a glimpse of how AI can be incorporated into clinical practice in radiology [27]. They showed that bone subtraction maps can help radiologists improve their efficiency and accuracy in detecting progressive MM on follow-up imaging.

### 4.3. Other Types of Studies

The other studies discussed in this paper used CT values in different ways to predict bone marrow infiltration [26,28]. While one study used cumulative CT values [26], the other tried to measure the amount of non-fatty bone marrow [28]. They both showed that these measures can be used to detect bone marrow infiltration levels in MM patients and, if used, can potentially reduce the number of bone marrow biopsies. Given the importance of explainability in clinical algorithms [38], these studies have the advantage of using simpler and more understandable methods for predictive tasks. This is compared to radiomics studies that use quantitative features that are hard to explain and difficult to correlate with the outcome in question by clinicians.

### 4.4. CLAIM Evaluation

The evaluation of the studies using the CLAIM checklist provides a clear picture of the areas that need improvement when conducting AI research in multiple myeloma. Researchers need to be more transparent regarding the methods they use. Some commonly overlooked components include particularly how they de-identified their data, how they initialized their model parameters, and a more detailed description of their sample, including how they would justify their sample size and how patients were included. Finally, while many studies provided performance metrics, they did not explore cases where their model had failed or tried to provide any explanations as to why these failures might happen. 

### 4.5. Recommendations for Future Directions

Given the fast pace at which AI research is changing, we expect to see a shift towards more advanced methods like deep learning models, generative AI, and large language models. In doing so, we believe that researchers need to consider a variety of factors, many of which are mentioned in the CLAIM criteria as well. First, we recommend that researchers use commonly used, widely accepted performance metrics (like AUROC, AUPRC, precision, recall, etc.) in a transparent way to enable comparison between different studies. We also recommend the use of explainable models or strategies to explain, for each case, how the model made its decision. This would not only increase the trustworthiness of their models but also make it easier for experts to evaluate them. Providing the code that was used to generate results and the data (if possible) is another step that is necessary for the evaluation of a research project. Other strategies include following the available guidelines when conducting AI research and using an external validation set. 

Another aspect of AI research on MM that is often overlooked is the transition of these models to actual healthcare settings. Studies that investigate how these tools can help the medical team improve care are scarce. Once a model is developed and validated using external datasets, researchers should think about how these models can transition to actual medical practice and design pilot studies that objectively evaluate this.

While the transition to care lags behind current research in the application of AI in multiple myeloma imaging, there is great potential for the implementation of these tools in practice. Risk prediction models based on radiomics or deep learning can help clinicians identify high-risk patients and personalize treatments accordingly. In addition, explainable AI models may be able to help researchers identify new potentially causal relationships and biomarkers that could be relevant in the management of patients. Finally, studies focusing on the segmentation or identification of lesions may be used both for screening patients for new lesions and as a first step for other AI pipelines (e.g., segmentation of bone marrow as a first step to generate radiomics features).

With the advent of new methods, AI is making its way into medical workflows. Given the huge amount of medical imaging data generated, radiology is one of the areas that is at the forefront of this change. In MM, we showed that research is underway to either make care easier by automating some tasks (ROI segmentation) or, even further, perform tasks that a human radiologist is not capable of by using complex algorithms (e.g., predicting the chance of relapse). In the future, we expect to see more studies that combine multiple AI tools to create a workflow that can provide valuable information to physicians. For example, an AI workflow might automatically detect all MM lesions in the body and guide radiologists in interpreting the image, thus reducing the time needed for the radiologist to perform this task (which is currently significant) and decreasing the chance of error. In addition, it will feed this data into another model that will provide estimated risk scores and personalized treatment suggestions that would add additional valuable information to the radiology report. Even further, we expect to see more multi-modal models that are going to use not only imaging data but also data from other sources to make more accurate predictions. These changes will make AI models valuable tools in a radiologist’s toolbox.

### 4.6. Limitations

The application of AI in MM imaging is still in its early stages. Hence the number of studies on this topic is few. However, we expect a surge in new studies in the following years. Another limitation of our work is that due to the diversity of research questions in the studies we reviewed and the variability of performance measures used, we could not compare the study findings. However, we tried to evaluate all studies using the CLAIM criteria to provide some means of comparison between the studies.

## 5. Conclusions

Based on current trends, we anticipate that radiology is going to play an even bigger role in the management of MM patients. In addition, the development of accurate segmentation algorithms can potentially lead to tools that can assist radiologists in their tasks by removing unnecessary details and highlighting more important imaging features. Supplementary algorithms can enrich radiologic reports with estimates of important features like the percentage of plasma cell infiltration and disease severity. Future research in this area could focus on using more advanced modeling strategies like deep learning, developing explainable predictive models, and implementing these models in clinical care.

## Figures and Tables

**Figure 1 diagnostics-13-03372-f001:**
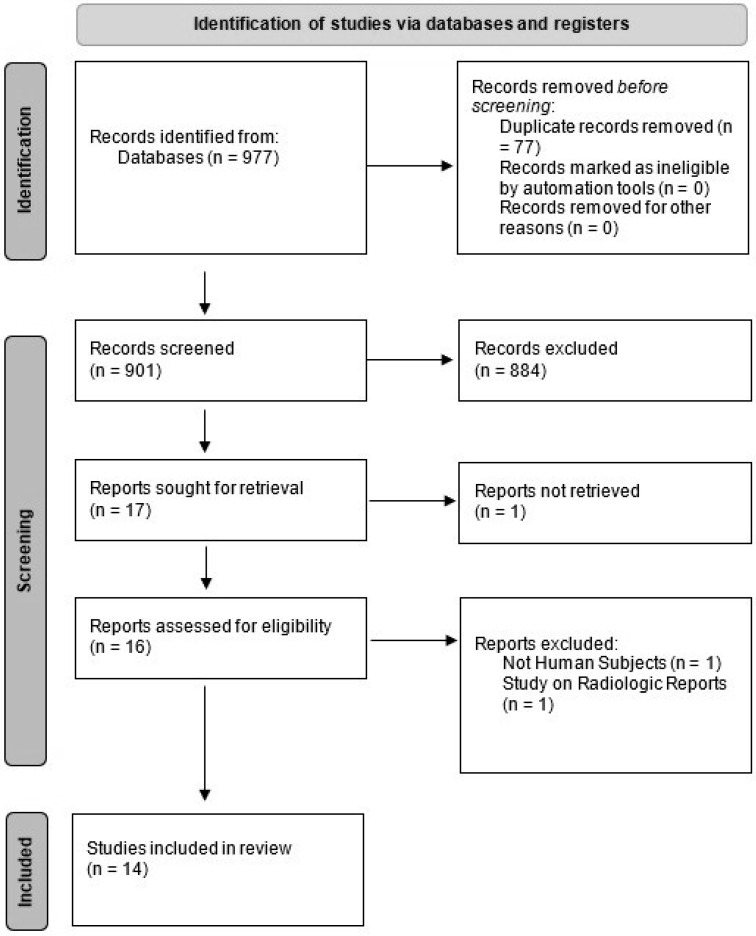
PRISMA flow chart of our study selection.

**Figure 2 diagnostics-13-03372-f002:**
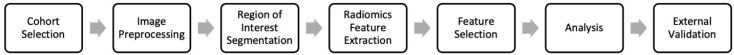
Typical radiomics workflow. The workflow includes selecting a cohort of patients eligible for the radiomics study, extracting the relevant imaging data, preprocessing the images, annotating the region of interest (e.g., bone lesions in MM), extracting the radiomics features using tools such as the Pyradiomics package, selecting the most relevant features, analysis, and finally external validation.

**Table 1 diagnostics-13-03372-t001:** An overview of the reviewed studies. For each article, the year of publication, the sample population, the type of imaging used, the type of input to the model, the gold standard, and the final objective of the project are presented.

Authors (Year)	N	Imaging	Input *	Segmentation	Feature Reduction	Model/Analysis	Standard Test	Objective
Fervers et al., (2021) [28]	35	CT	CT values	Pretrained CNN	N/A	Multivariate regression	N/A	Estimate BM infiltration
Fraenzel et al., (2011) Abstract [24]	14	CT	Image	Threshold model	N/A	Threshold segmentation and flood fill algorithm	N/A	Automatic bone lesion detection
Horger et al., (2017) [27]	188	CT	N/A	N/A	N/A	N/A	Lab results and gold standard CT	Monitoring new bone lesions
Li et al., (2021) [17]	121	MRI	Image	Manual segmentation	Univariate regression + Spearman’s correlation + LASSO	Nomogram	N/A	To predict overall survival in MM patients using radiomics and clinical features
Liu et al., (2021) [32]	50	MRI	Radiomics	Manual segmentation	ICC + LASSO	Logistic regression	FISH	Detection of high-risk cytogenic abnormality
Martínez-Martínez et al., (2016) [25]	127	CT	Image	Shape model positioning	N/A	Probabilistic density model	Imaging diagnosis	Bone marrow infiltration
Nishida et al., (2017) [26]	68	CT	Cumulative CT values	Threshold model	N/A	Pearson’s correlation	Histology	BM infiltration
Satoh et al., (2022) [29]	98	PET/CT	Image	Deep learning-based automatic organ segmentation	N/A	Histogram analysis	N/A	Establish a standard for BM FDG uptake
Schenone et al., (2021) [23]	33	CT	Clinical data + Image	Manual Segmentation	PCA, Pearson’s correlation	FCM, Extended version of HTF	Clinical diagnosis	Prognosis
Shi et al., (2018) Abstract [31]	12	PET/CT	Image	Manual segmentation	N/A	V-Net and W-Net	Manual segmentation	Computer-aided lesion detection
Takahashi et al., (2020) [30]	58	PET/CT	Standardized uptake values	Fully automated segmentation	N/A	Generalized estimating equation	Visual analysis	Standardizing VOI determination in PET/CT scans
Wennmann et al., (2021) Abstract [33]	66	MRI	Image	Manual segmentation	N/A	nnU-Net	Manual segmentation	BM segmentation
Wennmann et al., (2021) Abstract [35]	270	MRI	Radiomics	nn-Unet	N/A	Random Forest	Manual segmentation, biopsy	Estimate plasma cell infiltration
Xiong et al., (2021) [34]	107	MRI	Radiomics	Manual segmentation	ICC + LASSO	ANN	Clinical diagnosis	Differentiation of metastatic lesions from MM

N: sample size, BM: bone marrow, FDG: fludeoxyglucose, FCM: clustering fuzzy c-means, HTF: Hough transformation, MM: multiple myeloma, ICC: intraclass correlation, PCA: principal components analysis, ANN: artificial neural network, ICC: intra-class correlation coefficient, and VOI: volume of interest; * Input: data types used to make a model.

**Table 2 diagnostics-13-03372-t002:** An overview of the results and conclusions of the reviewed studies. Model performances are reported under the performance column to show the comparison of study findings.

Radiomics				
Authors (Year)	Methods	Performance	Conclusion	
Schenone et al., (2021) [23]	FCM, Extended version of HTF	Critical success index: 0.52 ± 0.1	Radiomic has the potential for the stratification of relapsed and non-relapsed MM patients.	
Xiong et al., (2021) [34]	LASSO + Artificial Neural Networks	MCC for ANN on T2WI: 0.605	Machine learning on radiomics features extracted from conventional MRI sequences can help differentiate newly diagnosed MM lesions from other cancer metastatic lesions in lumbar vertebrae.	
Li et al., (2021) [17]	LASSO + Nomogram	Radiomics nomogram c-index on validation cohort: 0.81 (95% CI: 0.70–0.92)	The development of a radiomics nomogram can help with the prediction of overall survival in MM patients.	
Liu et al., (2021) [32]	LASSO + Logistic Regression	Radiomics test AUC: 0.863 Combined test AUC: 0.870	Radiomics features in conventional MRI images of MM patients can help distinguish HRCAs and non-HRCAs.	
Wennmann et al., (2021) [35]	Random Forest	MAE: 14.3 compared to biopsy	The tool has an accuracy comparable to a radiologist in predicting plasma cell infiltration percentage.	
**Segmentation**				
**Authors (year)**	**ROI**	**Classifier**	**Performance**	**Conclusion**
Fraenzel et al., (2011) [24]	Medullary Cavity of Bones	Random Forest	Classification accuracy: Femur: 90%, Tibia: 79%, Fibula: 79%, Humerus: 93%, Radius: 69%, Ulna: 46%, Other: 99%	Classification of medullary cavities enables the identification of long bone structures in whole-body CT scans.
Shi et al., (2018) [31]	Bone Lesions	V-Net, W-Net	V-Net: Sensitivity: 71.1%, Specificity: 99.5%, W-Net: Sensitivity 73.5%, Specificity: 99.6%	W-Nets have superior performance compared to traditional ML models in the detection of bone lesions.
Takahashi et al., (2020) [30]	Bone	Threshold	SUVmean was correlated with visual assessment, OR: 10.52 (95% CI, 5.68–19.48), *p* < 0.0001	CT–based skeletal segmentation allows for the automated and therefore reproducible calculation of PET quantitative parameters of bone involvement in MM patients.
Wennmann et al., (2021) Abstract [33]	Bone Marrow	nn-Unet	Mean Dice scores ranging from 0.80 to 0.97	A Unet model can be used to segment out bone marrow structures of the body with high accuracy.
**Other**				
**Authors (year)**	**Measure**		**Performance**	**Conclusion**
Martínez-Martínez et al. (2016) [25]	Diagnosis accuracy		Infiltration vs. Healthy group: SVM, AUC: 0.996 ± 0.009; Infiltration vs. All other: k-NN, AUC: 0.894 ± 0.070	Classification based on features extracted from the probabilistic density model using k-NN, allowing differentiation of patients with infiltration from others.
Nishida et al., (2017) [26]	Correlation with clinical features		cCTv is correlated with higher R-ISS stage and serum or urine M-protein cCTv is inversely correlated with therapy and serum albumin	cCTv demonstrated a relationship with disease aggressiveness and had prognostic value.
Horger et al. (2017) [27]	Diagnosis accuracy		Performance of radiologist while using the tool: Sensitivity: 97.8%, Specificity: 96.7%, Accuracy: 97.7%	Accuracy was slightly increased and reading time was significantly reduced when using subtraction maps.
Satoh et al., (2022) [29]	N/A		SLUmean in men: 0.79 (95% CI 0.78–0.90), SLUmean in women: 0.75 (95% CI 0.74–0.76), SLUmean inversely correlated with age	A normal FDG uptake pattern was demonstrated by simplified FDG PET/CT bone marrow quantification.
Fervers et al., (2021) [28]	Diagnostic accuracy		AUROC for osteolytic lesions: 0.70 (95% CI, 0.49–0.90), AUROC for MM diagnosis: 0.71 (95% CI 0.54–0.89)	Automated, AI-supported attenuation assessment of the spine in DECT after VNCa is feasible to predict BM infiltration in MM.

FCM: clustering fuzzy c-means, HTM: Hough transformation, MM: multiple myeloma, SUV: standardized uptake volume, AUC: area under the curve, MCC: Mathews correlation coefficient, PCA: principal components analysis, ANN: artificial neural network, ML: machine learning, k-NN: K-nearest neighbors, cCTv: cumulative CT values, DECT: dual-energy computed tomography, VNCa: virtual non-calcium, SLU: standardized uptake volume corrected by lean body mass, and AUROC: area under the receiver operator curve.

**Table 3 diagnostics-13-03372-t003:** CLAIM criteria and the percentage of studies that were compliant with each component. For each criterion in CLAIM, an experienced AI researcher reviewed the full text of all articles and determined whether the criterion was fulfilled in the study. The numbers in the last column of this table show the percentage of articles that fulfilled this criterion.

Category	Subcategory	Item	Percentage of Articles That Were Compliant
Title/Abstract		Identification as a study of AI methodology, specifying the category of technology used (e.g., deep learning)	87
Title/Abstract		Structured summary of study design, methods, results, and conclusions	62
Introduction		Scientific and clinical background, including the intended use and clinical role of the AI approach	100
Introduction		Study objectives and hypotheses	100
Methods	Study design	Prospective or retrospective study	75
Methods	Study design	Study goals, such as model creation, exploratory study, feasibility study, and non-inferiority trial	87
Methods	Data	Data sources	100
Methods	Data	Eligibility criteria: how, where, and when potentially eligible participants or studies were identified. (e.g., symptoms, results from previous tests, inclusion in the registry, patient-care setting, location, and dates)	100
Methods	Data	Data pre-processing steps	75
Methods	Data	Selection of data subsets, if applicable	62
Methods	Data	Definitions of data elements, with references to common data elements	12
Methods	Data	De-identification methods	0
Methods	Data	How missing data were handled	50
Methods	Ground Truth	Definition of ground truth reference standard, in sufficient detail to allow replication	100
Methods	Ground Truth	Rationale for choosing the reference standard (if alternatives exist)	75
Methods	Ground Truth	Source of ground-truth annotations; qualifications and preparation of annotators	100
Methods	Ground Truth	Annotation tools	100
Methods	Ground Truth	Measurement of inter- and intrarater variability; methods to mitigate variability and/or resolve discrepancies	75
Methods	Data Partitions	Intended sample size and how it was determined	0
Methods	Data Partitions	How data were assigned to partitions; specify proportions	50
Methods	Data Partitions	Level at which partitions are disjoint (e.g., image, study, patient, and institution)	50
Methods	Model	Detailed description of the model, including inputs, outputs, all intermediate layers, and connections	87
Methods	Model	Software libraries, frameworks, and packages	62
Methods	Model	Initialization of model parameters (e.g., randomization, transfer learning)	0
Methods	Training	Details of the training approach, including data augmentation, hyperparameters, and number of models trained	50
Methods	Training	Method of selecting the final model	62
Methods	Training	Ensembling techniques, if applicable	0
Methods	Evaluation	Metrics of model performance	87
Methods	Evaluation	Statistical measures of significance and uncertainty (e.g., confidence intervals)	75
Methods	Evaluation	Robustness or sensitivity analysis	62
Methods	Evaluation	Methods for explainability or interpretability (e.g., saliency maps), and how they were validated	62
Methods	Evaluation	Validation or testing on external data	0
Results	Data	Flow of participants or cases, using a diagram to indicate inclusion and exclusion	37
Results	Data	Demographic and clinical characteristics of cases in each partition	62
Results	Model Performance	Performance metrics for optimal model(s) on all data partitions	87
Results	Model Performance	Estimates of diagnostic accuracy and their precision (such as 95% confidence intervals)	75
Results	Model Performance	Failure analysis of incorrectly classified cases	0
Discussion		Study limitations, including potential bias, statistical uncertainty, and generalizability	87
Discussion		Implications for practice, including the intended use and/or clinical role	100
Other Information		Registration number and name of registry	25
Other Information		Where the full study protocol can be accessed	12
Other Information		Sources of funding and other support; role of funders	100

## Data Availability

All the articles and data sources reviewed in this systematic review are publicly available online. Readers can access the full-text articles and associated data through the respective journal websites, databases, or repositories where they were originally published or hosted.

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
