# Peer review of "Current Status and Future of Artificial Intelligence in MM Imaging: A Systematic Review"

_diagnostics, 2023, doi:10.3390/diagnostics13213372_

Round 1

Reviewer 1 Report

Comments and Suggestions for Authors

Introduction:

The introduction effectively sets the stage by introducing multiple myeloma (MM) and the importance of artificial intelligence (AI) in medical imaging. It is engaging and gives a good background.

Methods:

The methods section provides a general overview of the approach taken for the systematic review and details regarding the search strategy, inclusion, and exclusion criteria. I would be useful to mention if any limitations encountered.

Results:

The results section is thorough and effectively presents the findings. It successfully summarizes the characteristics of the included studies, providing insights into the various applications of AI in MM imaging.

CLAIM Checklist:

The application of the CLAIM checklist is a valuable addition, highlighting transparency aspects. It might be useful to elaborate a bit more on what each compliance aspect entails and how it was evaluated in the reviewed studies.

Discussion:

The discussion section provides a thorough analysis of trends in AI research related to MM imaging and outlines the potential impact of AI in this context. To further improve this section:

Consider offering recommendations or insights for future research directions based on the findings. For instance, suggest areas where additional AI applications may be particularly promising or where further methodological improvements are needed.

Discuss the clinical implications of the reviewed AI applications. How might these technologies benefit patients and healthcare providers in the real-world clinical setting? Providing practical insights can enhance the relevance of the review.

In summary, the article provides a strong foundation in discussing AI in MM imaging. To enhance its completeness and clarity, consider adding a clear research question, providing more specific methodological details, elaborating on the CLAIM checklist evaluation, and offering recommendations in the Discussion. With these improvements, the article can be even more impactful and informative.

Author Response

Introduction:

  1. The introduction effectively sets the stage by introducing multiple myeloma (MM) and the importance of artificial intelligence (AI) in medical imaging. It is engaging and gives a good background.

Author Response:

Thank you for your comment.

Methods:

  1. The methods section provides a general overview of the approach taken for the systematic review and details regarding the search strategy, inclusion, and exclusion criteria. I would be useful to mention if any limitations encountered.

Author Response:

Thank you for your comment.

Author Action:

On page 14 at the end of discussion, we added a section on limitations:

“The application of AI in MM imaging is still at the early stages. Hence the number of studies on the topic is small. However, we expect a surge of new studies in the following years. Another limitation of our work is that due to the diversity of research questions in the studies we reviewed, and the variability of performance measures used, we could not compare study findings together. However, instead we tried to evaluate all studies using the CLAIM criteria to provide some means of comparison between the studies.”

Results:

  1. The results section is thorough and effectively presents the findings. It successfully summarizes the characteristics of the included studies, providing insights into the various applications of AI in MM imaging.

Author Response:

Thank you for your comment.

CLAIM Checklist:

  1. The application of the CLAIM checklist is a valuable addition, highlighting transparency aspects. It might be useful to elaborate a bit more on what each compliance aspect entails and how it was evaluated in the reviewed studies.

Author Response:

Thank you for your comment. We agree that we should clarify how each criterion was evaluated.

Author Action:

On page 4 in the study evaluation section, we added the following explanation:

“A physician with 4 years of experience in medical AI research reviewed each paper to determine compliance to each of the criteria. Compliance was defined based on the descriptions provided in the original CLAIM paper. For example, for the “How missing data were handled” criteria, an article was considered compliant if they stated that they did not have any missing data or if they described the strategy they used to deal with missing data.”

In addition, although we could only mention one example in the text, on Table 3, we described each criterion separately with a short description on what it would mean to be compliant.

Discussion:

The discussion section provides a thorough analysis of trends in AI research related to MM imaging and outlines the potential impact of AI in this context. To further improve this section:

  1. Consider offering recommendations or insights for future research directions based on the findings. For instance, suggest areas where additional AI applications may be particularly promising or where further methodological improvements are needed.

Author Response:

Thank you for your comment. We agree that adding a section to the discussion regarding recommendations or insights can improve the paper.

Author Action:

On page 13, we added a section titled “Recommendations for future directions”. Then we added the following two paragraphs:

” Given the fast pace at which AI research is changing, we expect to see a shift towards more advanced methods like deep learning models, generative AI and large language models. In doing this, we believe that researchers need to consider a variety of factors, some of which are mentioned in the CLAIM criteria as well. First, we would recommend researchers to use commonly used, widely accepted performance metrics (like AUROC, AUPRC, precision, recall, etc.) in a transparent way to enable comparison between different studies. We also recommend the use of explainable models or strategies to explain, for each case, how the model made its decision. This would not only increase the trustworthiness of their models, but it would also make it easier for experts to evaluate them. Providing the code that was used to generate results and the data (if possible) is another step that is necessary for evaluation of a research project. Other strategies include following the available guidelines when doing AI research and having an external validation set.

Another aspect of AI research in MM that is often overlooked is the transition of these models to actual healthcare settings. Studies that investigate how these tools can help the medical team improve care are sparse. Once a model is developed a validated using ex-ternal datasets, researchers should think about how these models can transition to actual medical practice and design pilot studies that objectively evaluate this.”

  1. Discuss the clinical implications of the reviewed AI applications. How might these technologies benefit patients and healthcare providers in the real-world clinical setting? Providing practical insights can enhance the relevance of the review.

Author Response:

Thank you for your comment. We agree with your comment.

Author Action:

On page 14 we added the following paragraph about the clinical AI applications:

“While transition to care lags behind current research in the application of AI in multiple myeloma imaging, there is a great potential for these tools to be implemented in practice. Risk prediction models based on radiomics or deep learning could help clinicians identify high risk patients and personalize treatments accordingly. In addition, explainable AI models may be able to help researchers identify new causal relationships and biomarkers that could be relevant in management of patients. Finally, works focusing on segmentation or identification of lesions may be used both for screening of patients for new lesions and as a first step for other AI pipelines.”

In summary, the article provides a strong foundation in discussing AI in MM imaging. To enhance its completeness and clarity, consider adding a clear research question, providing more specific methodological details, elaborating on the CLAIM checklist evaluation, and offering recommendations in the Discussion. With these improvements, the article can be even more impactful and informative.

Reviewer 2 Report

Comments and Suggestions for Authors

This paper presents a review of artificial intelligence methods in multiple myeloma imaging. I have following comments toward your manuscript.

1. The review is interesting, however, the structure of the paper needs improvement.

2. I think categorizing methods is a better choice.

3. Figures should be improved, i.e. Fig.2 is not clear, Fig.3 is too large, etc.

4. I do not understand Table 3.

5. The scope of method application  may be more discussed.

6. The document should be better organized. 

Comments on the Quality of English Language

Minor editing

Author Response

This paper presents a review of artificial intelligence methods in multiple myeloma imaging. I have following comments toward your manuscript.

  1. The review is interesting, however, the structure of the paper needs improvement.

Author Response:

Thank you for your comment. We agree with your comment.

Author Action:

We added new headings to the discussion section and adjusted the structure based on categorizing the methods as mentioned in comment 2. These included specific parts to discuss radiomics, segmentation and other types of studies separately in both the results section and the discussion section. We also added headings and sections related to CLAIM criteria evaluation, future directions and limitations.

  1. I think categorizing methods is a better choice.

Author Response:

Thank you for your comment. We agree with your comment.

Author Action:

We categorized the rest of the parts of the article based on the method that was used in the studies (radiomics, segmentation, others)

  1. Figures should be improved, i.e. Fig.2 is not clear, Fig.3 is too large, etc.

Author Response:

Thank you for your comment. We agree with your comment.

Author Action:

Figure 2 was removed as it did not add a lot of information and was causing confusion. We reduced the size of figure 3. In addition, we added the following information to the caption for figure 3 (Now figure 2 as the original figure 2 was removed): “The workflow includes selecting a cohort of patients eligible for the radiomics study, extracting the relevant imaging data, preprocessing the images, annotation of the region of interest (e.g., bone lesions in MM), extraction of the radiomics features using tools such as the Pyradiomics package, selection of the most relevant features, analysis, and finally external validation.”

  1. I do not understand Table 3.

Author Response:

Thank you for your comment.

Author Action:

We have added the following description to the caption for table 3 to make it clearer: “For each criterion in CLAIM, an experienced AI researcher reviewed the full text of all articles and determined if the criterion was fulfilled in the study. The numbers in the last column of this table shows what percentage of articles that fulfilled that criterion.”

  1. The scope of method application  may be more discussed.

Author Response:

Thank you for your comment. We agree with your comment.

Author Action:

On page 14, we added the following paragraph that describes the potential clinical/research applications of the methods described:

“While transition to care lags behind current research in the application of AI in multiple myeloma imaging, there is a great potential for these tools to be implemented in practice. Risk prediction models based on radiomics, or deep learning could help clinicians identify high risk patients and personalize treatments accordingly. In addition, explainable AI models may be able to help researchers identify new causal relationships and biomarkers that could be relevant in the management of patients. Finally, works focusing on segmentation or identification of lesions may be used both for screening of patients for new lesions and as a first step for other AI pipelines.”

  1. The document should be better organized. 

Author Response:

Thank you for your comment. We agree with your comment.

Author Action:

We added new headings to the discussion section and adjusted the structure based on categorizing the methods as mentioned in comment 2. These included specific parts to discuss radiomics, segmentation and other types of studies separately in both the results section and the discussion section. We also added some headings and sections related to future directions and limitations.

Reviewer 3 Report

Comments and Suggestions for Authors

Thanks for the possibility to review your work. I have some revisions I would kindly ask authors to address.

1 It is recommended that the method section in the abstract provide more specific details to help readers understand the research process.

It is recommended to briefly and clearly explain the overview and clinical manifestations of multiple myeloma (MM), so that readers can directly understand the basic information of the disease in the first paragraph of the Introduction section.

It is recommended to provide a more specific explanation of the extracted content and significance in the Data Extraction section, so that readers can better understand the specific usage scenarios and potential limitations of the data.

It is recommended to provide a more detailed description of the characteristics and differences in case selection, imaging modalities, and study design when referring to the included studies.

It is recommended to provide a more detailed introduction to the specific content and purpose of each step in the first paragraph of the Radiomics study section.

It is recommended to use tables for data comparison in the Radiomics research section.

It is recommended to provide an overview of the first method of the second major category in the Discussion section and other subsequent analyses.

8 It is recommended to describe the improvement and development trends of this study in the Conclusion section.

Comments on the Quality of English Language

The article is relatively smooth overall, with only a few other areas where sentence usage needs to be corrected

Author Response

Thanks for the possibility to review your work. I have some revisions I would kindly ask authors to address.

  1. It is recommended that the method section in the abstract provide more specific details to help readers understand the research process.

Author Response:

Thank you for your comment. We agree with your comment that more detail may be added.

Author Action:

On page 1, in the methods section of the abstract, we added a few more sentences. The methods section of the abstract now reads: “This study was performed based on PRISMA guidelines. Three main concepts were used in the search algorithm, including “artificial intelligence” in “radiologic examinations” of patients with “multiple myeloma”. The algorithm was used to search in PubMed, Embase, and Web of Science databases. Articles were screened based on the inclusion and exclusion criteria. At the end, we used the Checklist for Artificial Intelligence in Medical Imaging (CLAIM) criteria to evaluate the manuscripts. We provided the percentage of studies that were compliant with each criterion as a measure of the quality of AI research in MM.”

  1. It is recommended to briefly and clearly explain the overview and clinical manifestations of multiple myeloma (MM), so that readers can directly understand the basic information of the disease in the first paragraph of the Introduction section.

 Author Response:

Thank you for your comment. We agree with your comment.

Author Action:

On page 1, in the first paragraph of the introduction, we have provided an overview of MM. We added the following information to complement this paragraph: “5 Other symptoms of the disease include weight loss, fatigue or general weakness, pares-thesia, hepatomegaly, splenomegaly and fever.”

  1. It is recommended to provide a more specific explanation of the extracted content and significance in the Data Extraction section, so that readers can better understand the specific usage scenarios and potential limitations of the data.

Thank you for your comment. We agree with your comment.

Author Action:

On page 4, we adjusted the text for the first paragraph to include more detail about the extracted content:

“Authors’ names, year, descriptive data of all studies including sample size, study de-sign, imaging modality, techniques, parameter, reference standard, and the subjective of each study were extracted. The following characteristic data were also obtained if they were provided:  feature reduction strategies that are often used in radiomics studies to prevent overfitting, the analysis tool that was used for the project including methods like ridge regression, LASSO, XGBoost and deep learning, the diagnostic performance measures provided including area under the receiver operator curve (AUROC), accuracy, sensitivity, specificity, number of readers, portion of sample size they have used for training a model, conclusion, and pros and cons of each paper.”

  1. It is recommended to provide a more detailed description of the characteristics and differences in case selection, imaging modalities, and study design when referring to the included studies.

Thank you for your comment. We agree with your comment that this needs more clarification.

Author Action:

On page 4, table 1, we have provided an overview of each study, their sample population and modeling strategies. In addition, we altered the table caption to highlight this: “For each article, the year of publication, sample population, the type of imaging used, the type of input to the model, the gold standard, and the final objective of the project is presented.”

  1. It is recommended to provide a more detailed introduction to the specific content and purpose of each step in the first paragraph of the Radiomics study section.

Thank you for your comment. We agree with your comment.

Author Action:

On page 7, under the radiomics section, we altered the first paragraph to add more detail about the radiomics pipeline:

“Radiomics was one of the main focuses of studies looking into AI applications in MM imaging. Radiomics is a set of quantitative features extracted from medical images that have proven useful in predicting disease features and outcomes.36 A typical pipeline for a radiomics study includes acquiring a set of images (with the same scanner and protocol), normalization of those images using bias correction and normalization methods like z-score normalization in addition to resampling of the images so that each pixel for all images have the same size and images are uniform, ROI segmentation which could be manually done by a radiologist or automatic using AI models to delineate the actual ROI (usually bone marrow or bone lesions in MM), calculating radiomics features, often done using packages like Pyradiomics, selecting the most relevant ones using feature selection strategies like LASSO that select only the most informative features, and finally doing the analysis using machine learning techniques like ridge regression, decision trees or deep learning.”

  1. It is recommended to use tables for data comparison in the Radiomics research section.

Thank you for your comment. We agree that we should clarify this part. We have included the results of the radiomics studies in table 2 under the performance column.

Author Action:

In page 5 in the caption of table 2, we provided more details on how findings could be compared using this table.

  1. It is recommended to provide an overview of the first method of the second major category in the Discussion section and other subsequent analyses.

Thank you for your comment. We agree with your comment.

Author Action:

On page 13 we reorganized the text and under the “ROI segmentation in MM” part. In addition, we added the following description to better explain the methodology: “Segmentation in radiology is the process of categorizing each pixel or voxel in a medical image to a specific anatomical or pathological class, such as identifying and labeling different structures or regions within the image, for diagnostic or analytical purposes.“ In addition, we added the following example to better describe rule based segmentation: ”An example could be segmentation of bone by selecting regions of the image with Hounsfield values equal or higher than that of bone”

  1. It is recommended to describe the improvement and development trends of this study in the Conclusion section.

Thank you for your comment. We agree with your comment.

Author Action:

On page 15, we added the following sentences to the conclusion section: “Future research in this area could be focused on using more advanced modeling strategies like deep learning, developing explainable predictive models and the implementation of these models in clinical care.”

Round 2

Reviewer 2 Report

Comments and Suggestions for Authors

I thank the authors for the manuscript improvement. 

Comments on the Quality of English Language

Most aspects have been enhanced. It can be accepted.

Reviewer 3 Report

Comments and Suggestions for Authors

In the manuscript, the authors have made effective modifications to my recommendations. In my opinion, this manuscript can be accepted in present form.

Comments on the Quality of English Language

The overall use of English language in the article is relatively smooth, with only a small portion of grammar that needs to be corrected and optimized